# Practical Experience of Sensitivity Analysis: Comparing Six Methods, on Three Hydrological Models, with Three Performance Criteria

**Anqi Wang** [1],* **and Dimitri P. Solomatine** [2,3,4],*

[1]   College of Hydrology and Water Resources, Hohai University, NO.1 Xikang Road, Nangjing 210098, China
[2]   Chair Group of Hydroinformatics, IHE Delft Institute for Water Education, Westvest 7, 2611AX Delft, The Netherlands
[3]   Water Resources Section, Delft University of Technology, Mekelweg 5, 2628CD Delft, The Netherlands
[4]   Water Problems Institute, Russian Academy of Sciences, Gubkina 3, Moscow 117735, Russia
*   Correspondence: wanganqi0718@163.com (A.W.); d.p.solomatine@tudelft.nl (D.P.S.)

**Abstract:** Currently, practically no modeling study is expected to be carried out without some form of Sensitivity Analysis (SA). At the same time, there is a large number of various methods and it is not always easy for practitioners to choose one. The aim of this paper is to briefly review main classes of SA methods, and to present the results of the practical comparative analysis of applying them. Six different global SA methods: Sobol, eFAST (extended Fourier Amplitude Sensitivity Test), Morris, LH-OAT, RSA (Regionalized Sensitivity Analysis), and PAWN are tested on three conceptual rainfall-runoff models with varying complexity: (GR4J, Hymod, and HBV) applied to the case study of Bagmati basin (Nepal). The methods are compared with respect to effectiveness, efficiency, and convergence. A practical framework of selecting and using the SA methods is presented. The result shows that, first of all, all the six SA methods are effective. Morris and LH-OAT methods are the most efficient methods in computing SI and ranking. eFAST performs better than Sobol, and thus it can be seen as its viable alternative for Sobol. PAWN and RSA methods have issues of instability, which we think are due to the ways Cumulative Distribution Functions (CDFs) are built, and using Kolmogorov–Smirnov statistics to compute Sensitivity Indices. All the methods require sufficient number of runs to reach convergence. Difference in efficiency of different methods is an inevitable consequence of the differences in the underlying principles. For SA of hydrological models, it is recommended to apply the presented practical framework assuming the use of several methods, and to explicitly take into account the constraints of effectiveness, efficiency (including convergence), ease of use, and availability of software.

**Keywords:** Global Sensitivity Analysis; hydrological model; bootstrapping resample

## 1. Introduction

Water management makes wide use of hydrological models. Models are uncertain, and it mainly comes from the error of gathering input data, e.g., rainfall and evapotranspiration, parameters of the model, and the model structure itself. Nowadays, the interests to Uncertainty Analysis (UA) methods and procedures have grown considerably. The study of the UA will not only improve the credibility of the model itself, but will also be conductive to decision making under uncertainty.

UA can be defined differently—see e.g., [1–5], but in general it gives a qualitative or quantitative assessment of the uncertainty in the model results. The results are typically expressed as a graph showing the spread and ensemble of values or a distribution, as probabilistic flood maps, etc.

Due to the complexity and non-linear nature of hydrological models, it is hard to use analytical methods to study the uncertainty of hydrological models. Therefore, non-intrusive, sampling-based methods are commonly used, generally referred to as Monte Carlo Simulation (MCS), which can be seen as the simulation of a system that encloses stochastic or uncertain components. It can be easily implemented and is model independent and dimension independent.

The Monte Carlo method can be expressed as "the use of random sampling as a tool to produce observations on which statistical inference can be performed to extract information about a system" [6]. MCS starts firstly with generating $n$ samples of some variables $X$. These variables can be external model inputs, initial model conditions, or model parameters. For each such realization, simulation of the model $Y = f(X)$ is carried out to obtain $n$ sets of output (could be either time series or single value), which statistics are analysed.

Often, however, yet another concept is employed to analyze the impact of uncertainties on modelling results, Sensitivity Analysis (SA), which is ideologically close to UA. It can be defined as the study of "how the uncertainty in the output of a model (numerical or otherwise) can be apportioned to different sources of uncertainty in the model input" [4]. (One may notice that this definition is not comprehensive, since uncertainty not only comes from model inputs but also from parameters, so for this reason, we will use the term "factor" instead of "model input".) The main aim here is to identify the degree with which changes in various factors (manifesting the corresponding uncertainty) influence a change in model output. SA should be seen as a standard step in any modelling study, and there is plenty of literature on SA published during the last 40–50 years, but still various updates and improvements of SA techniques are proposed regularly (see e.g., [7–9]).

SA is often implemented before model parameterization (calibration). On one hand, for conceptual rainfall-runoff models, the parameters cannot be gathered from field measurement and implementing, and SA can help to find out the most influential parameters to reduce the cost of calibration time. On the other hand, for distributed hydrological models, whose parameters can be gathered from the field, SA can help to target the most important parameters, on which more resources can be put to ensure their higher accuracy. (It should be noted that there is a certain danger and even a methodological flaw in conducting SA of parameters before model calibration: it is not yet really known what the optimal parameter vector is, and hence it is possible that sensitivity is investigated considering non-feasible parameters values. Therefore, it would be advisable to carry out at least some initial calibration before turning to SA).

The main difference between UA and SA lies in the fact that SA tries to explicitly apportion the uncertainty of the output to the different factors, and does not aim at studying the uncertainty of model outputs in detail. Therefore, SA can help to target the sources of the model output uncertainty due to that in inputs or parameters, whereas UA provides a more general and often more detailed and rigorous account of model uncertainty, which is, however, computationally more demanding.

Saltelli et al. [4] formulates the three main specific purposes of SA:

- Factor Prioritization (FP): *ranking* the factor in terms of their relative sensitivity;
- Factor Fixing (FF), or *screening*: determining the factors are influential or not to the output uncertainty;
- Factor Mapping (FM): given specific output values or ranges, locating the regions in the factor space that produces them.
- In this study, we only focus on ranking and screening.

SA is typically categorized into Local Sensitivity Analysis (LSA) and Global Sensitivity Analysis (GSA). LSA concentrates on the sensitivity of factors at particular points in the factor space, for example, around the vector of the calibrated parameters. GSA, on the other hand, assesses the sensitivity of the factor through the whole factor space. By design, LSA is simpler and faster.

A simplest expression of local sensitivity is the first-order partial derivatives of output to the factors. Define a model $y = f(x)$, where $y$ is the output of the model; $x$ is factor of the model. The sensitivity of the factor ($S$) is defined as:

$$S_i = \left| \frac{\Delta y_i}{\Delta x_i} \right| \qquad (1)$$

where $i$ is the i-th factor of the model. (Note that in many studies instead of model output $y$ the model error is used, e.g., Root Mean Squared Error or Mean Absolute Error.) Higher value of $S_i$ indicates higher sensitivity of the factor. Such measure of sensitivity is often called Sensitivity Index (SI). Figure 1 shows the expression of sensitivity of a model with two parameters (factors).

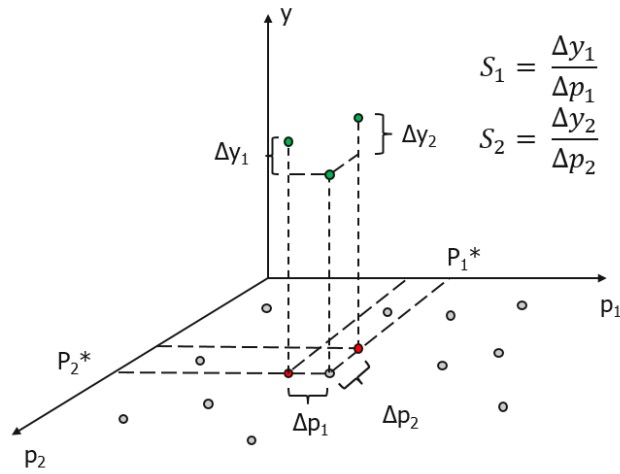

**Figure 1.** Graphical expression of Sensitivity Analysis.

One may want also to explore sensitivity for more factor vectors, by randomly sampling several points in the whole factor space (or its subspace), and obtaining $S_i$ for each sample point. After that the results can be aggregated (e.g., in simplest case calculating the mean value of these $S_i$), providing an estimate of the global sensitivity of the model.

Such Global Sensitivity Analysis methods can be classified into Generalized Sensitivity Analysis method, variance-based methods, GLS (globally aggregated measure of local sensitivities) methods, density-based methods, and meta-modelling methods. Different methods are based on different theories and principles, and as a result, have different efficiencies. Which method is the best to use is always an issue to discuss in the field. There are various studies comparing different SA methods. In the study of Tang et al. [10], four SA methods have been analyzed and compared on SAC-SMA model (Sacramento Soil Moisture Accounting model) coupled with SNOW-17. The results of the study show that the choice of SA methods has great impacts on the parameter sensitivity of the model. Pappenberger et al. [11] tested five SA methods on a flood inundation model (HEC-RAS). It is demonstrated that different methods result in different ranking of factors, thus solid conclusions about the sensitivity of the factors are impossible to draw. Gan et al. [12] have evaluated the effectiveness and efficiency of ten widely used SA methods on SAC-SMA model. The result demonstrates that qualitative SA methods (which provide heuristic score to intuitively represent the relative sensitivity factors) are more efficient than quantitative SA methods (computing the impact of the parameter on the total variance of model output), whereas quantitative SA methods are more robust and accurate. Song [13], Razavi and Gupta [14], and Pianosi et al. [15] gave very useful systematic reviews of SA concepts, methods, and framework, respectively, with suggestions on how to choose SA methods. However, these suggestions are only made based on dissimilar studies and literature reviews, without a comprehensive comparison of SA methods applied to one case study.

With respect to sample-based SA methods, the coverage of the factor space is the key point in SA accuracy: with small samples the SA results are imprecise. Also, different sets of sample may give

dissimilar SA results. In other words, there is also uncertainty in SA (in fact, the same can be said about UA). In order to deal with this issue, convergence of the Sensitivity Indices should be studied.

In spite of many reviews and comparison of SA methods that have been carried out, there are not too many studies that investigate the convergence and uncertainty of the SA results. Yang [16] assessed the convergence of Sensitivity Indices for five different Global Sensitivity Analysis methods using Central Limit Theory (CLT) and bootstrap techniques. In her study, the estimates of mean and Confidence Interval (CI) are plotted against increasing base sample size for each method. Once there is no significant fluctuation in the values, the convergence is reached. Sarrazin et al. [17] proposed a methodology to study the convergence of Sensitivity Indices, ranking, and screening. They have defined quantitative criteria for the convergence of Sensitivity Indices, ranking, and screening, and tested the methodology on the three widely-used GSA methods applied to three hydrological models. Our research adds to these two mentioned studies, with more methods and models.

Another aspect worth attention is the choice of SA method(s). Most of the studies concerning SA could not draw firm conclusions about how to choose the best SA method (and this is understandable, since there are many ways to define what is the "best" one). Also, the uncertainties associated with the procedures of SA are not investigated much.

Objectives of this study are as follows. The first objective of the study is to additionally test and compare the widely used classic SA methods as well as the SA methods developed recently (e.g., PAWN, Pianosi and Wagener [9]) in the aspects of efficiency, effectiveness, and convergence. The second objective is to give suggestions on how to choose SA methods for various hydrological (or hydraulic models) based on their computational cost, robustness, and easiness of implementation. The third objective of the study is present a structured sequence of practical steps to implement SA and UA, which can be also seen as a framework of sensitivity and uncertainty analysis of hydrological models. The presented framework we use is quite close to the guidelines and frameworks published earlier (e.g., [4,13,15,18], which conceptually are also close), and our aim here is not to replace but to complement them, giving special attention to the performance analysis. Clearly, each individual SA study has its specifics so it is hardly possible to have a unified framework or procedure that would fit all possible requirements. Each researcher or practitioner would have a choice of various approaches, principles, and components to combine and follow in SA, depending on the require depth of analysis and available resources.

The structure of the paper is as follows. Section 2 gives a detailed introduction and description of Global Sensitivity Analysis methods. Section 3 presents the methodology and case study of this study to evaluate GSA methods. The results of the study are shown in Section 4 and followed by discussions in Section 5. Finally, conclusions are drawn in Section 6.

## 2. Global Sensitivity Analysis Methods

This section is in no way a detailed presentation of the methods, but rather a brief introduction to the techniques compared in this study, and to some issues of their implementation. For comprehensive reviews, please refer to Song et al. [13], and Pianosi et al. [15], and for a relatively recent interesting insight into the SA problem, to Razavi and Gupta [7,8,14].

### 2.1. Classification of Global Sensitivity Analysis (GSA) Methods

2.1.1. Generalized (Regionalized) Sensitivity Analysis, and Other Density-Based Methods

Generalized Sensitivity Analysis method (also referred to as Regionalized SA) has gained popularity in environmental and water-related research in the end of 1970s, especially after the papers by Spear and Hornberger [19,20] and, to some extent Whitehead and Young [21], and it is also worth checking the earlier work by Spear [22]. This approach was positioned as a Monte Carlo framework used for "probabilistic calibration" which aimed at finding regions in parameter space leading sets of "behavioral" (good) and "non-behavioral" (bad) models (which point at the regions of critical

uncertainty), rather than aiming at finding one "best" model. Simulation results are split into these two groups based on their performance (e.g., model error), the Cumulative Distribution Function (CDF) of each factor is generated for each group, and their difference is analyzed. Typically, the Kolmogorov–Smirnov statistic [23] is used to compute the discrepancy between these CDFs. GSA allows for identifying the regions of the model parameter space in which parameters have the significant effect on the model behaviour. One can see also that GSA, being based on the Mote Carlo framework and using statistical analysis of outputs, can be also seen as a representative of UA.

One of the drawbacks of RSA is that the results are influenced by the selection of different thresholds, which undermines its objectivity. To resolve this problem, Wagener et al. [24] presented an extension of this method. The parameter sets are grouped into ten groups instead of two, based on the model performance. They are sorted from best to worst, in which the first group produces the best 10% results (e.g., the results with least 10% model error), the second group produces the best 10–20% results, and so on. Empirical CDFs of the parameters are also plotted for each group, if the curves are concentrated or overlapped, the parameters are not sensitive, and vice versa. The quantification of SI can be achieved by computing Kolomogorov–Smirnov statistic, which is the maximum vertical distance between two eCDFs (empirical CDF). For detailed description and implementation of the method, please refer to Jakeman et al. [24] and Wagener et al. [25].

RSA belongs to a wider group of methods which explore PDFs or CDFs of the output. Sensitivity is measured by the comparison of unconditional PDF derived from purely random samples and conditional PDF derived when prescribing one factor. Entropy-based sensitivity measures [26–28] and the δ-sensitivity measure [29,30] are implementations of this concept.

The production of empirical PDFs is a crucial step in most of the density-based methods. However, the derivation of empirical PDFs is either too simple, so that the results may not be accurate, or may be computationally too complex to implement (for reasons of such complexity, see [9,28]). Recently, Pianosi and Wagener [9] proposed a novel method called PAWN that partly overcomes this difficulty. The key idea of PAWN method is to compare the unconditional CDF of output with conditional CDFs of output which prescribe one parameter at a fixed value (the conditioning value), while others vary randomly.

### 2.1.2. Variance-Based Methods

The density-based methods aim at analyzing the output distributions, but often it is enough to concentrate on some moments only, e.g., on variance. Variance-based methods are today perhaps the most popular approaches for SA. The underlying assumption of variance-based methods is that the sensitivity can be measured by the contribution of the factor's variance (the contribution of the factor itself, or interactions with two or more factors) to the variance of the output. The biggest advantage of a variance-based method is that it can compute the main effect and higher-order effect of factors respectively, and make it distinguishable which factors have high influence on the output on their own, and which factors have high interaction with others.

It is normally unrealistic to analytically compute the Sensitivity Index because of the complexity of hydrological models. Instead, Sobol proposed an efficient sample-based approach to compute first and total-order Sensitivity Indices—called the Sobol method—which is perhaps the most popular variance-based SA method [31]. A detailed description of the method and its implementation can be also found in Saltelli et al. [4].

Though the result of the Sobol method is robust, often considered as benchmark run for study, it is computationally expensive, requiring a large number of base samples. Another popular approach to numerically compute variance-based Sensitivity Indices is the Fourier Amplitude Sensitivity Test (FAST), presented by Cukier et al. [32]. The key idea of FAST is applying the ergodic theorem to transform the $n$-dimension integral to one-dimension integral. Saltelli and Bolado [33] provide a detailed description of principles and procedures for implementation of the method. One of the drawbacks of

the FAST method is that it can only compute the main effect. However, an improved version of FAST, which is extended FAST (eFAST) [34], can compute first and total order Sensitivity Indices.

### 2.1.3. Globally Aggregated Measure of Local Sensitivities (GLS) Method

As mentioned in Section 1, the globally aggregated measure of local sensitivities methods use average value of SA measures (e.g., first-order derivative) at each local sample points in the factor space as Sensitivity Index for each factor.

Morris [35] proposed an approach which he referred to as Elementary Effects Test (EET) to compute the sensitivity. It is also called Morris Screening method. Its modification was proposed by Campolongo et al. [36]. Its principle concept is to use the mean and standard deviations of the gradients of each sample as the measure of the overall effect and interaction effect of each factor across the $p$ level factor space. Morris Screening is a simple but effective method, widely used for screening in hydrological modelling. A more detailed description of the method can be found in Saltelli et al. [4].

Since sampling is time-consuming, it is reasonable to use economical techniques for it, and e.g., van Griensven et al. [37] employed Latin Hypercube Sampling, followed by assessments of the local error derivatives at each point "one at a time" (OAT), which they named LH-OAT method. The Sensitivity Index of each factor is obtained by averaging the derivatives of all perturbed samples.

All GLS methods conceptually are quite simple and their reported implementations typically do not require large number of runs. However, Razavi and Gupta [14] have pointed out that they may suffer from scale issue, that is, the selection of the step size may influence the results due to the complexity of response surface of the model.

### 2.2. Use of Meta-Modelling to Reduce Running Times of Global Sensitivity Analysis (GSA) Methods

Sampling used in SA requires considerable computational time, for complex models prohibitively long. The basic idea of meta-modelling is to substitute the original model (and hence its response function linking factors and model output) with a simpler function or a model. This substitution is typically done by using statistical or machine (statistical) learning techniques, and employing methods of the so-called experimental design for generating data by the model runs to be used for training the meta-model. SA is carried out using the meta-model, and for this, the variance-based method is mostly used.

Techniques used for this purpose include Radial-basis function network (RBF) [38], multivariate adaptive regression splines (MARS) [39], support vector machine (SVM) [40], Gaussian processes (GP) [41] and treed Gaussian processes (TGP) [42]. The advantage of meta-modelling is that by simplification of the original complex model, the overall running time is considerably decreased; the trade-off is a possible loss of accuracy.

## 3. Methodology and Experimental Set-Up

### 3.1. Methodology for Evaluating SA Methods

#### 3.1.1. The Three Evaluation Criteria

Different SA methods have different concepts and principles behind them, and, accordingly, the Sensitivity Indices may have different meanings and metrics. However, it would be logical to try to follow the general principles behind any method for a model (method) evaluation, i.e., effectiveness and efficiency. The evaluation of SA methods' *effectiveness* is aimed at finding out whether the relative Sensitivity Indices, ranking, and screening of parameters make sense and can be used in SA. *Efficiency* of SA methods is assessed by how fast (in terms of computational time) they provide the result: the lesser number of model runs is required, the more efficient the method is. Therefore, the evaluation of SA methods efficiency is to figure out the minimum number of runs required for each SA method to get satisfactory results—and it is not always clear and explicitly defined what "satisfactory" actually

means. Due to the fact that sampling is employed, there is always uncertainty in the SA results, and in the values of the Sensitivity Indices calculated depend on the sample size. In order to take into account the uncertain nature of SA results, the *convergence* of the SA results should be studied, and this forms the last aspect of the evaluation of SA methods. It is worth noting that the concepts of efficiency and convergence are closely related, because more efficient SA methods require fewer model runs and hence converge faster. The (subtle) difference is in the following: efficiency aims at assessing how fast the (reliable) SA results become available, while analysis of convergence goes further, to quantification of uncertainty of SA results (its distribution, or at least the width of the confidence interval).

### 3.1.2. Evaluation of Effectiveness

The result of SA is not an absolute one and nobody can say what the "correct answer" is. Unlike assessing the accuracy of a hydrological model, which can be compared with the observation values, for sensitivity there are no "observations" to be compared with. To start somewhere, we will initially randomly sample a large number (say, 10,000) parameter (factor) vectors and run the model for each of them. The RMSE (Root Mean Square Error) of the model output will be plotted against parameter values as a scatter plot which will provide a rough image of the sensitivity of each parameter. The preliminary assessment of the sensitivities of each parameter will be treated as a reference. Then all considered SA methods will be run, and their results will be compared with the reference to assess their performance. Effectiveness will be evaluated on the three aspects: Sensitivity Indices values, ranking, and screening.

We realize that constructing a reference this way provides quite a rough estimation of sensitivity, and this is an inevitable limitation. Therefore, the results of all the methods will be taken into account, compared, and analyzed to see the differences and similarities between them and not only with the reference.

### 3.1.3. Evaluation of Efficiency

For each method, one benchmark test will be run with a considerable size of the base sample set of 10,000. Different base sample sizes will be set for each SA method, to be compared with the results of its benchmark run. From the results, the minimum base sample size will be found for each SA method to ensure the effective results in terms of Sensitivity Indices stability and factors ranking.

### 3.1.4. Evaluation of Convergence

Convergence of Sensitivity Indices will be analyzed by calculating 95% Confidence Intervals and mean value for increasing sample sizes. To increase the confidence of estimates, bootstrapping (see e.g., [43]) will be used as well. The following procedure will be employed (adapted from Yang [16]):

1.　Generate N samples of parameters as the base sample set.
2.　The N base samples are re-sampled B times with replacement, and for each replica, the Sensitivity Indices are computed, producing B Sensitivity Indices to construct the distribution of them.

From this sampling distribution, statistics of the Sensitivity Indices distribution (95% Confidence Interval and mean value) is calculated to quantify uncertainty.

### 3.2. Case Study

The presented methods have been tested on two case studies: Dapoling–Wangjiaba catchment in China, and the Bagmati catchment located in central Nepal. Due to data limitations issues, not all experiments with the first case have been finalized, so it is not reported here, and is left for the future publications.

The Bagmati catchment covers an area of approximately 3700 km$^2$ (see Figure 2). The altitude of the region varies from 2913 m in the Kathmandu Valley, to Terai Plain, where it reaches the Ganges River in India, with an altitude of 57 m. The Bagmati River has an extension of about 195 km, flowing

from Shivapuri to the Ganges River in the south. In this study, focus is put on the part of the basin that drains to the Pandheradobhan station, with an area of 2900 km$^2$ and river length of 134 km. Solomatine and Shrestha [44] have used this case study in their paper on using machine learning for predicting uncertainty of hydrological models.

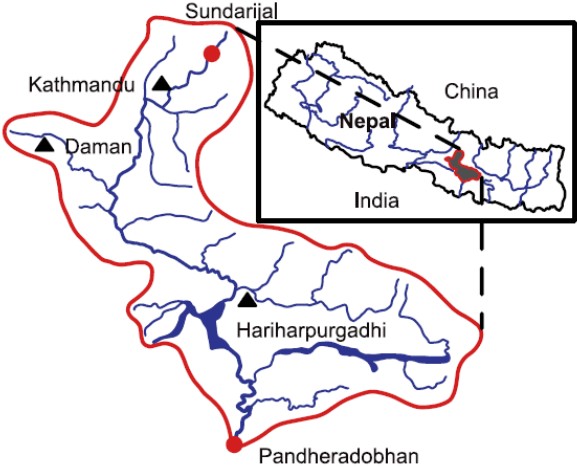

**Figure 2.** Location of the Bagmati catchment. Triangles denote the rainfall stations and circles denote the discharge gauging stations.

In this study, daily precipitation and air temperature from Kathmandu, Hariharpurgadhi, and Daman station and daily discharge in Pandheradobhan station from 1 March 1991 to 31 December 1995 are used. The daily average precipitation was assessed using Theissen polygon method and the potential evapotranspiration is calculated by the modified Penman method recommended by the Food and Agriculture Organization—FAO [45]. The hydrograph is shown in Figure 3.

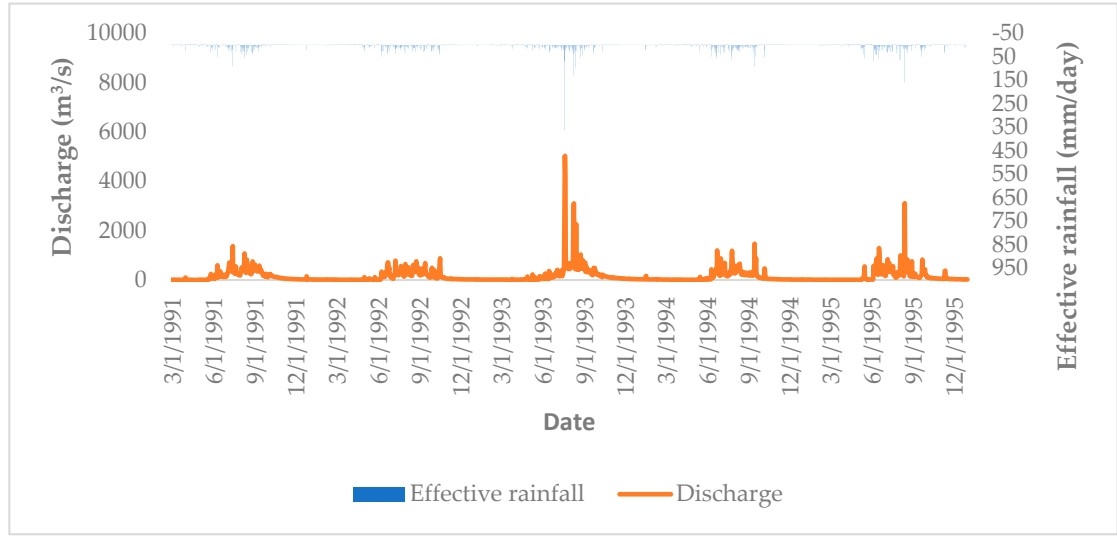

**Figure 3.** Hydrograph of the Bagmati Catchment from 1 March 1991 to 31 December 1995.

*3.3. Test Model*

The SA will be tested on three conceptual rainfall-runoff models: GR4J, Hymod, and HBV, with increasing complexity and the parameters number.

The modèle du Génie Rural (Agricultural Engineering Model) à 4 paramètres Journalier (4 parameters Daily, GR4J) was developed by Perrin et al. [46]. It uses daily precipitation and evapotranspiration as input to simulate the runoff discharge. The model structure assumes that after neutralization of precipitation

by evapotranspiration, a portion of net rainfall goes to production store, where percolation takes place. The leakage flow, together with the remaining part of the net rainfall, go to routing store, where they are split into two parts and routed by two unit hydrographs. After exchanging with groundwater, the total runoff is generated by adding these two parts. The structure of GR4J model is shown in Figure 4. The four parameters with their meaning and ranges are shown in Table 1.

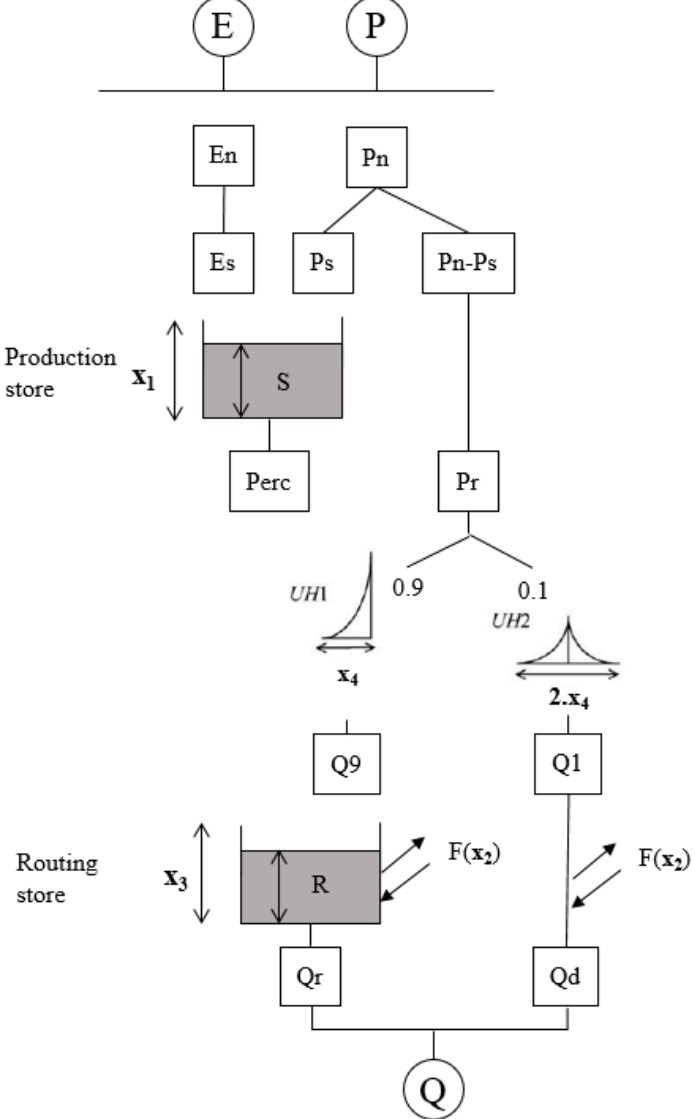

**Figure 4.** Diagram of GR4J model.

**Table 1.** Description and ranges of parameters in GR4J model.

| Parameter | Description | Unit | Lower Bound | Upper Bound |
|---|---|---|---|---|
| $X_1$ | Production store: Storage of rainfall in the surface of soil | mm | 1 | 1500 |
| $X_2$ | Groundwater exchange coefficient: a function of groundwater exchange which influence routing store | mm | −10 | 5 |
| $X_3$ | Routing storage: amount of water can be stored in soil porous | mm | 1 | 500 |
| $X_4$ | Time peak: the time when the ordinate peak of flood hydrograph is created | day | 0.5 | 4 |

The Hymod model, first introduced by Boyle [47] and presented in Wagener et al. [24], has been used quite widely for rainfall-runoff modelling because of its simplicity. It consists of a simple rainfall excess model with two parameters and a routing module with three parameters. In the rainfall excess model, the soil moisture storage capacity is assumed to be variable, described by a distribution function. The routing module contains two sets of parallel linear reservoirs. Three identical linear reservoirs account for the fast runoff component and a single linear reservoir accounts for the slow runoff component. The structure of Hymod model is shown in Figure 5. The name, meaning, and ranges of the parameters are shown in Table 2.

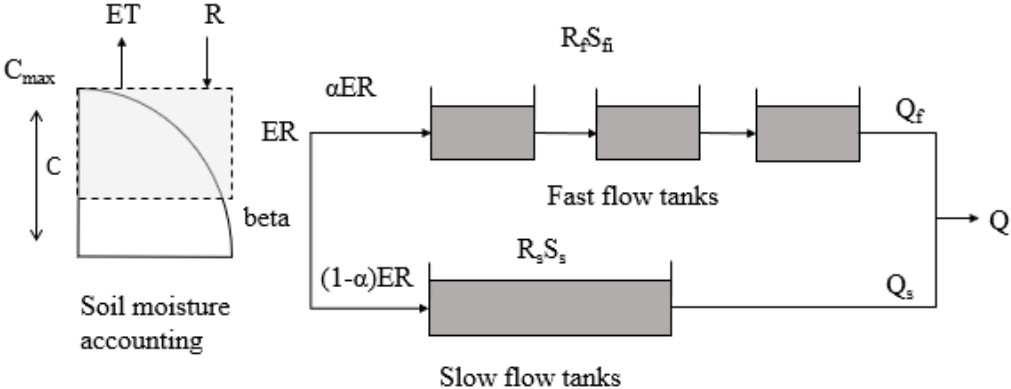

**Figure 5.** Diagram of Hymod model.

**Table 2.** Description and ranges of parameters in Hymod model.

| Parameter | Description | Unit | Lower Bound | Upper Bound |
| --- | --- | --- | --- | --- |
| SM | Maximum soil moisture | mm | 0 | 400 |
| BETA | Exponential parameter in soil routing | - | 0 | 2 |
| ALFA | Partitioning factor | - | 0 | 1 |
| RS | Slow reservoir outflow coefficient | - | 0 | 0.1 |
| RF | Fast reservoir outflow coefficient | - | 0.1 | 1 |

The HBV (Hidrologiska Bryåns Vattenbalansaldevning) model is a conceptual rainfall-runoff model widely used in Europe. It was developed by the Swedish Meteorological and Hydrological Institute [48] and then promoted by Lindström et al. [49] to become the HBV-96 model. In this study, a simplified version of the HBV-96 model is used. It consists of the three main modules, which is characterized as tank respectively, with 13 parameters: four of the parameters are related to snow accumulation and melt module, four with soil moisture accounting module, and five with river routing and response module. For river routing and response module, two runoff reservoirs are included. The upper non-linear reservoir accounts for the quick flow and the lower linear reservoir accounts for the base flow. Since there is little snowfall in the applied case study, the snow accumulation and melt module are excluded, so only nine parameters will be analyzed. The structure of HBV-96 model is shown in Figure 6. The name, meaning, and ranges of the parameters are shown in Table 3.

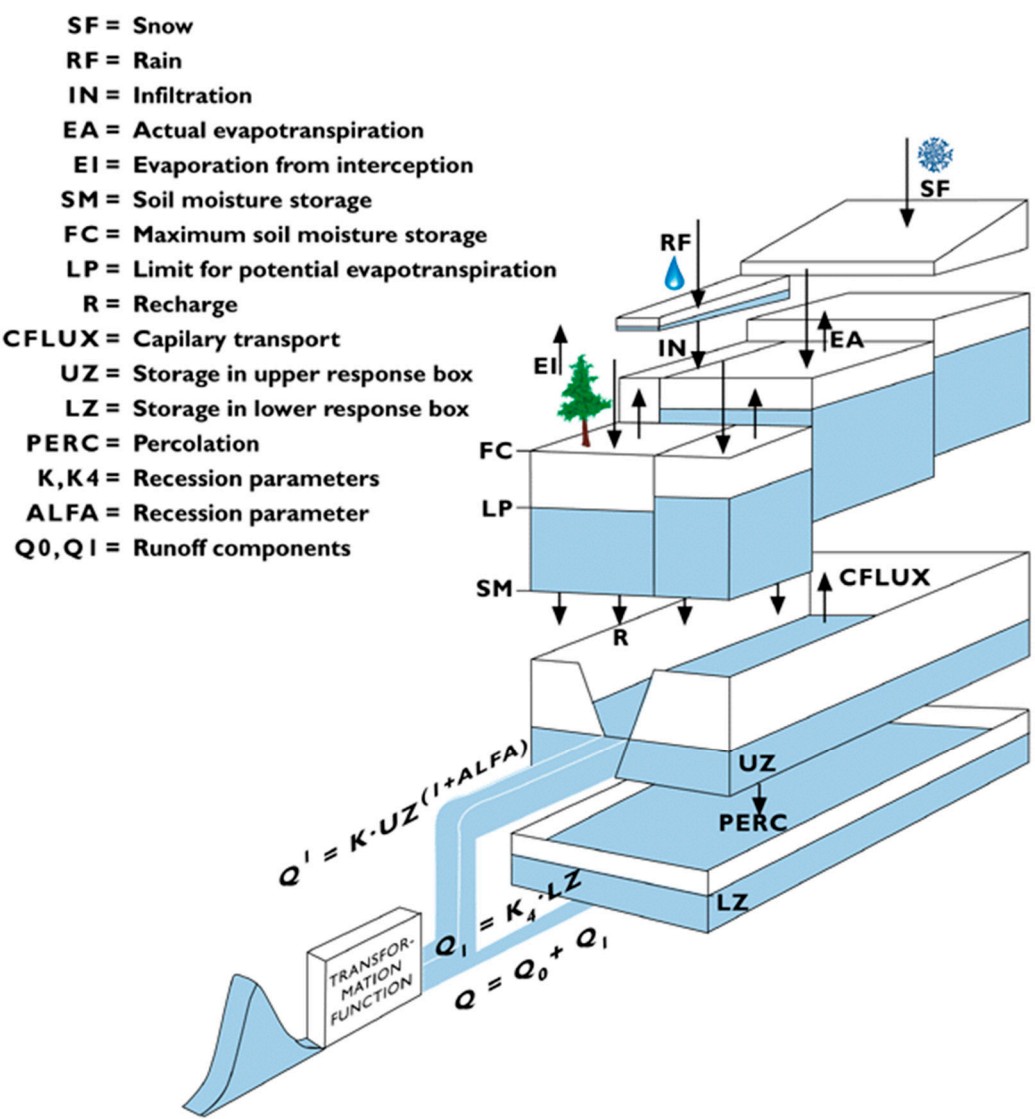

SF = Snow
RF = Rain
IN = Infiltration
EA = Actual evapotranspiration
EI = Evaporation from interception
SM = Soil moisture storage
FC = Maximum soil moisture storage
LP = Limit for potential evapotranspiration
R = Recharge
CFLUX = Capilary transport
UZ = Storage in upper response box
LZ = Storage in lower response box
PERC = Percolation
K,K4 = Recession parameters
ALFA = Recession parameter
Q0,QI = Runoff components

**Figure 6.** Diagram of Hidrologiska Bryåns Vattenbalansaldevning (HBV) model [50].

**Table 3.** Description and ranges of parameters in HBV model.

| Parameter | Description | Unit | Lower Bound | Upper Bound |
|---|---|---|---|---|
| FC | Maximum soil moisture content | mm | 50 | 500 |
| LP | Limit for potential evapotranspiration | - | 0.3 | 1 |
| ALFA | Response box parameter | - | 0 | 4 |
| BETA | Exponential parameter in soil moisture | - | 1 | 6 |
| K | Recession coefficient for upper tank | mm/d | 0.05 | 0.5 |
| K4 | Recession coefficient for lower tank | mm/d | 0.01 | 0.3 |
| PERC | Percolation from upper to lower tank | mm/d | 0 | 8 |
| CFLUX | Maximum value of capillary flow | mm/d | 0 | 1 |
| MAXBAS | Transfer function parameter | d | 1 | 3 |

### 3.4. Experimental Set-Up

The experimental set-up is presented in Table 4. The evaluation is done on six SA methods: Sobol, eFAST, Morris, LH-OAT, RSA, and PAWN. All software is implemented in MATLAB. For Sobol method, eFAST. Morris screening, RSA, and LH-OAT, the codes are constructed by the first author. For PAWN method, the codes from the SAFE toolbox [51] are used. In the present study, we follow a widely adopted approach when instead of studying the sensitivity of the model directly, the sensitivity of the

model error (deviation from observations) is analyzed instead. For the model error, we use the Root Mean Squared Error ($E_{RMSE}$):

$$E_{RMSE} = \sqrt{\frac{1}{n}\sum_{t=1}^{n}\left(Q_{sim}^{t} - Q_{obs}^{t}\right)^{2}} \tag{2}$$

where $Q_{sim}^{t}$ is the simulated model output at time step $t$, $Q_{obs}^{t}$ is the observation value at time step $t$, $n$ is the number of time steps. To avoid the influences of model initial states, the first three months (90 time steps) are excluded when computing $E_{RMSE}$.

One fact that needs to be pointed out is that base sample size is not equal to the number of model runs. Base sample size is the number of parameter sets sampled, while the actual number of model runs is normally larger than it. The number of model runs is determined by number of parameters, base sample size, and the SA method applied. The range of the parameters for sampling are shown in tables in Section 3.3, and the uniform distribution is assumed for each parameter.

Due to the characteristics of FAST sampling in eFast method, bootstrapping resample is not applicable, and evaluation of convergence will not be done for eFAST method. The resample size for other SA methods for evaluation is 100.

**Table 4.** Experimental set-up for evaluation of Sensitivity Analysis (SA) methods.

| Method | Measure | Sampling Method | Required Number of Runs | Parameters within the Method | Benchmark Run | Number of Base Samples for Evaluation |
|---|---|---|---|---|---|---|
| Sobol | Sobol total-order index | LHS | $(k + 2) \times N$ | - | $N = 10,000$ | $N = 100/200/300/500/1000/2000/3000/5000$ |
| eFAST | FAST total-order index | Fourier Amplitude Sensitivity Test (FAST) sampling | $K \times N$ | $M_s = 4$<br>$N_{cs} = 1$ | $N = 10,000$ | $N = 100/200/300/500/1000/2000/3000/5000$ |
| Morris | Modified mean of Effect Elementary | Morris one at a time | $(k + 1) \times N$ | $p = 32$<br>$\Delta = 0.5161$ | $N = 10000$ | $N = 100/200/300/500/1000/2000/3000/5000$ |
| LH-OAT | Effect S | LHS | $(k + 1) \times N$ | $\Delta = 0.05$ | $N = 10000$ | $N = 100/200/300/500/1000/2000/3000/5000$ |
| RSA | Mean of KS statistics | LHS | $N$ | - | $N = k \times 10000$ | $N = 100/200/300/500/1000/2000/3000/5000$ |
| PAWN | Max of KS statistics | LHS | $N_u + k \times n \times N_c$ | - | $N_u = 500$<br>$n = 40$<br>$N_c = 250$ | $[N_u, n, N_c] =$<br>$[30,10,10]/[50,10,20]/[100,15,20]/[100,20,25]/$<br>$[200,25,40]/[200,25,80]/[200,30,100]/[500,50,100]$ |

Notes: $k$ is the number of parameters; $N$ is the base sample size, $M_s$ is the number of higher harmonics to be considered; $N_{cs}$ is the number of search curves; $N_u$ is the number of samples for constructing unconditional CDFs; $n$ is the number of conditioning values for each parameter, $N_c$ is the number of samples for constructing conditional Cumulative Distribution Functions (CDFs). By the "benchmark run" we understand an experiment (the run), long enough to ensure convergence and an accurate estimate of sensitivity. For the detailed explanation of the parameters within each method please refer to the literature referred in Section 2.

## 4. Results

### 4.1. Preliminary Assessment of Sensitivity

The model was run 10,000 times; the scatter plots of the $E_{RMSE}$ against parameters for the three models are shown in Figures 7–9. From the scatter plot, the relative sensitivity of the parameter can be seen from the randomness of its distribution (i.e., proximity to the uniform distribution). The more randomly the RMSEs are distributed, the less sensitive the parameter is.

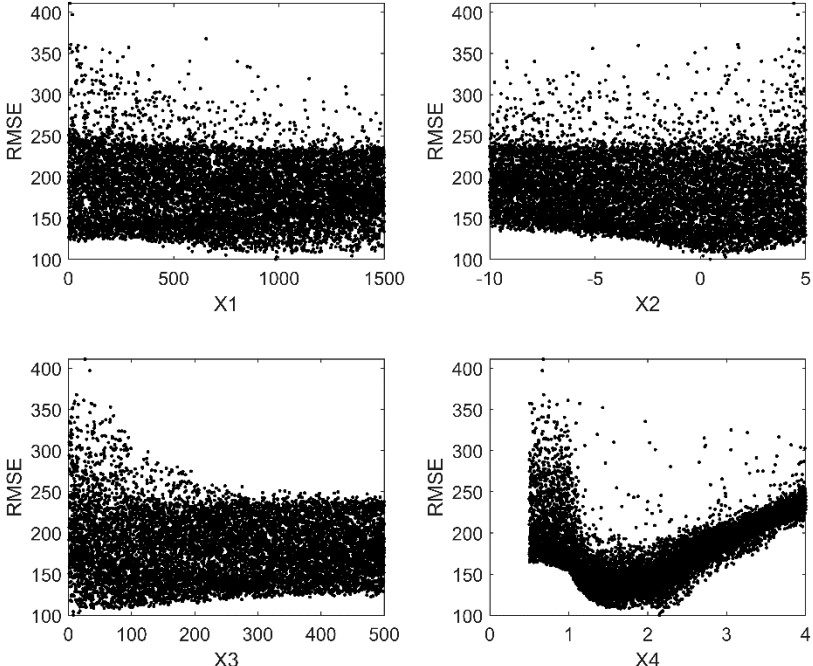

**Figure 7.** Scatter plot of Root Mean Square Error (RMSE) against parameter values with 10,000 runs for GR4J model.

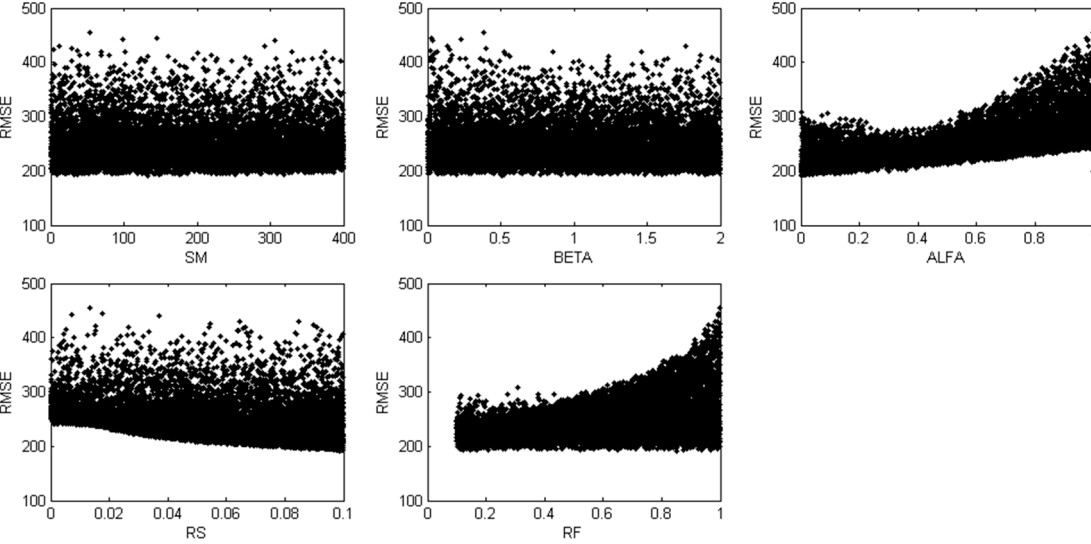

**Figure 8.** Scatter plot of RMSE against parameter values with 10,000 runs for Hymod model.

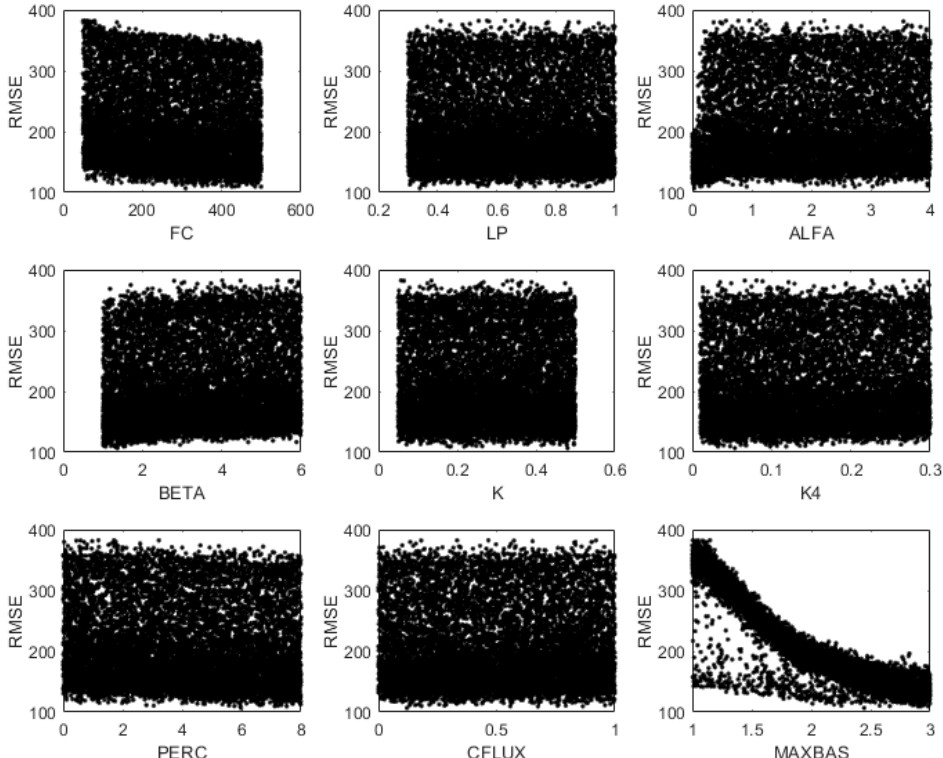

**Figure 9.** Scatter plot of RMSE against parameter values with 10,000 runs for HBV model.

In the GR4J model (Figure 7) X4 is shown to be the most influential parameter, followed by X3 and X2, while X1 seems to be of little influence. X4 is the time when the ordinate peak of flood hydrograph is created, which actually determines the shape of the hydrograph, so it is no surprise it appears to the most influential parameter in the model. X1 is the storage of rainfall in the soil surface, which does not affect the routing process too much, thus it is the least sensitive parameter.

In Hymod model (Figure 8), ALFA, RS, and RF have high influence on RMSE. SM and BETA, however, seem to be non-influential. This is understandable, because ALFA, RS, and RF control the fast and slow pathway in flow-routing module which is more important in determining flows, while SM and BETA only account for soil moisture routine which is less important.

In Figure 9, it can be seen that in HBV model, MAXBAS is obviously the most influential parameter; FC, ALFA, BETA, and PERC also show a certain degree of sensitivity; LP, K, K4, and CFLUX are non-influential. The reason is that MAXBAS is the (routing) transfer function parameter which controls the shape of the hydrograph.

*4.2. Effectiveness*

Figure 10 shows the results of benchmark runs of each method for three models, and one may see the following:

1.  all the methods identify the same set of sensitive parameters (X3 and X4 for GR4J, ALFA, RS, and RF for Hymod, MAXBAS for HBV);
2.  for less influential or non-influential parameters, different methods show relatively large discrepancy in results;
3.  the results of Sobol and eFAST are close, and it is also so for Morris and LH-OAT, RSA and PAWN, which indicates that the methods of the same category have similar results. This is due to the reason that both Sobol and FAST are variance-based methods, they all calculate the contribution of the variance to the output. Both Morris and LH-OAT compute the first-order partial derivatives

of the output. Similarly, RSA and PAWN use empirical CDFs and KS statistics to quantify the sensitivity. These groups of methods share the same principle;

4.  comparatively, the results of RSA and PAWN are always quite different from other methods. There may be two reasons: firstly, the generation of empirical CDFs may be inaccurate; secondly, the use of KS statistics to compute Sensitivity Index in both methods may also bring inaccuracy into the results (sensitivity to sampling) because KS statistics takes into account only the maximum difference between CDFs;

5.  ranking of parameters for the three models by different SA methods has many differences, but they are quite close in identifying sensitive and insensitive parameters, which means they are effective in screening.

In general, it can be said that all six methods are effective in computing SI. The results of RSA and (to a smaller extent) PAWN are to be treated with care because they use the (sensitive) KS statistics based only on the maximum difference in CDFs between the behavioral and non-behavioral models' sets.

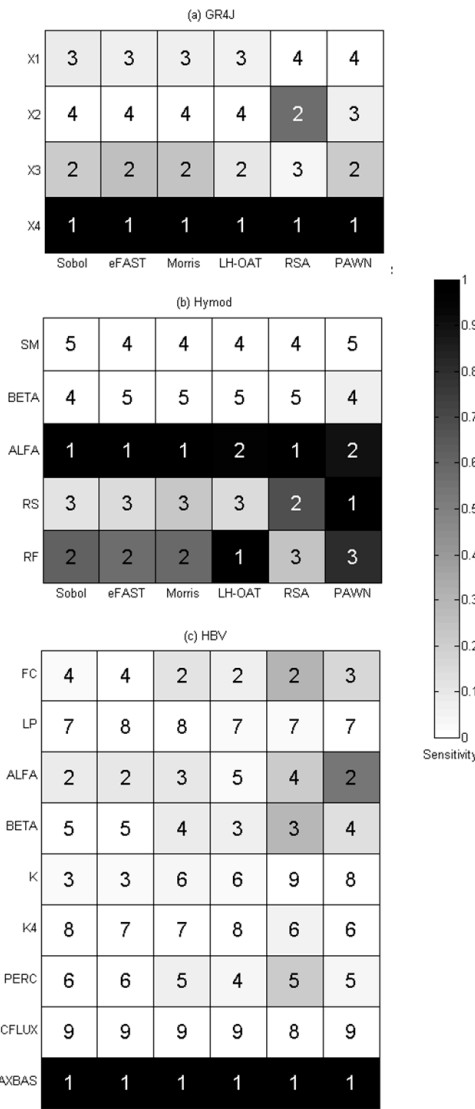

**Figure 10.** Sensitivity Indices (normalized) of six Sensitivity Analysis (SA) methods with benchmark run for GR4J (**a**), Hymod (**b**) and HBV (**c**) model, the number in the grid indicates the rank of the parameter within each SA method.

*4.3. Efficiency*

Figure 11 demonstrates the results of six SA methods for the three models for different number of runs. The minimum number of runs needed to get stable ranking of the parameters can be found in Table 5. As can be seen, among all the methods, Morris and LH-OAT converge quickly and are very stable across all numbers of run: they can get reliable SI and ranking at a very small number of runs (100 base sample for each model for both methods). eFAST is also quite stable, and it can get reliable results after approximately 300 base sample size. Comparing eFAST with Sobol method, it can be concluded that eFAST is more stable and more reliable (note also that at some point Sobol method results even in negative SI). RSA and PAWN are not very efficient, for the reason already stated in the previous section. RSA performs better than PAWN, especially for GR4J and Hymod model, for it can get stable SI at early runs. For PAWN method, the minimum number of runs to obtain reliable results is larger than for the other methods, and the reason is that it needs sufficiently large number of samples to create smooth eCDFs. Besides this, the sample size of the conditioning values will affect the conditional eCDFs. It also needs sufficient number of samples of conditioning values to cover the factor space well: this results in high computational cost since for each conditioning value k*Nc runs of the model are needed.

For all methods, it can be seen that with the increase of the model complexity and number of parameters, the results of SA become less stable. Especially for HBV model, except MAXBAS, all other parameters seem to be of similar sensitivity, therefore there is a considerable fluctuation in results. This can also be seen in GR4J and Hymod in which parameters have similar sensitivity (X1 and X3 for GR4J, SM, and BETA for Hymod).

*4.4. Convergence*

Figure 12 presents the estimates of the mean and the 95% Confidence Interval of all SA methods for three models with different number of runs. Overall, with increasing number of runs, the width of CI become narrower and have less and less variation. There are still differences in the width of CI and speed of convergence between the methods. It can be seen that Morris, LH-OAT, and RSA converge well already at early runs, and the width of CI are quite narrow across all runs. PAWN method converges comparatively slower and the width of CI is also wider. Sobol method is slowest, especially at small number of runs. The upper and lower bound of SI significantly exceed the range 0 to 1, which is quite unacceptable.

For all methods, similar conclusions as in efficiency can be drawn that with the increase of the model complexity and number of parameters, the uncertainty of SA also goes up. This increase of uncertainty also results in unstable results when the sensitivities of the parameters are close as shown in results of efficiency.

From the results shown above, it is proven that all six methods are effective in calculating Sensitivity Indices, screening, and ranking. Their efficiencies, however, differ. The minimum number of runs for computing Sensitivity Indices, ranking, and reaching convergence with each method are presented in Table 5.

In general, it takes many more runs to reach convergence, but many less runs are sufficient to obtain reliable ranking of the parameters. The Sobol method requires a large number of runs to be stable and reach convergence, which is very inefficient. Same as the variance-based method, eFAST method is much more efficient and stable. It is a good alternative for Sobol method with high efficiency. Morris and LH-OAT are also quite efficient and can provide results of ranking after relatively small number of runs. Also, the uncertainties of the values of Sensitivity Indices are not so high, and especially they are good at ranking and screening. The density-based methods, however, need sufficient number of runs to produce reliable eCDFs, and thus the efficiency is not so high. Furthermore, using KS statistics to compute Sensitivity Indices may be problematic for some types of distributions. Comparing RSA with PAWN, one can see that RSA performs better, especially for ranking, however, due to its design, it provides less detailed analysis of sensitivity.

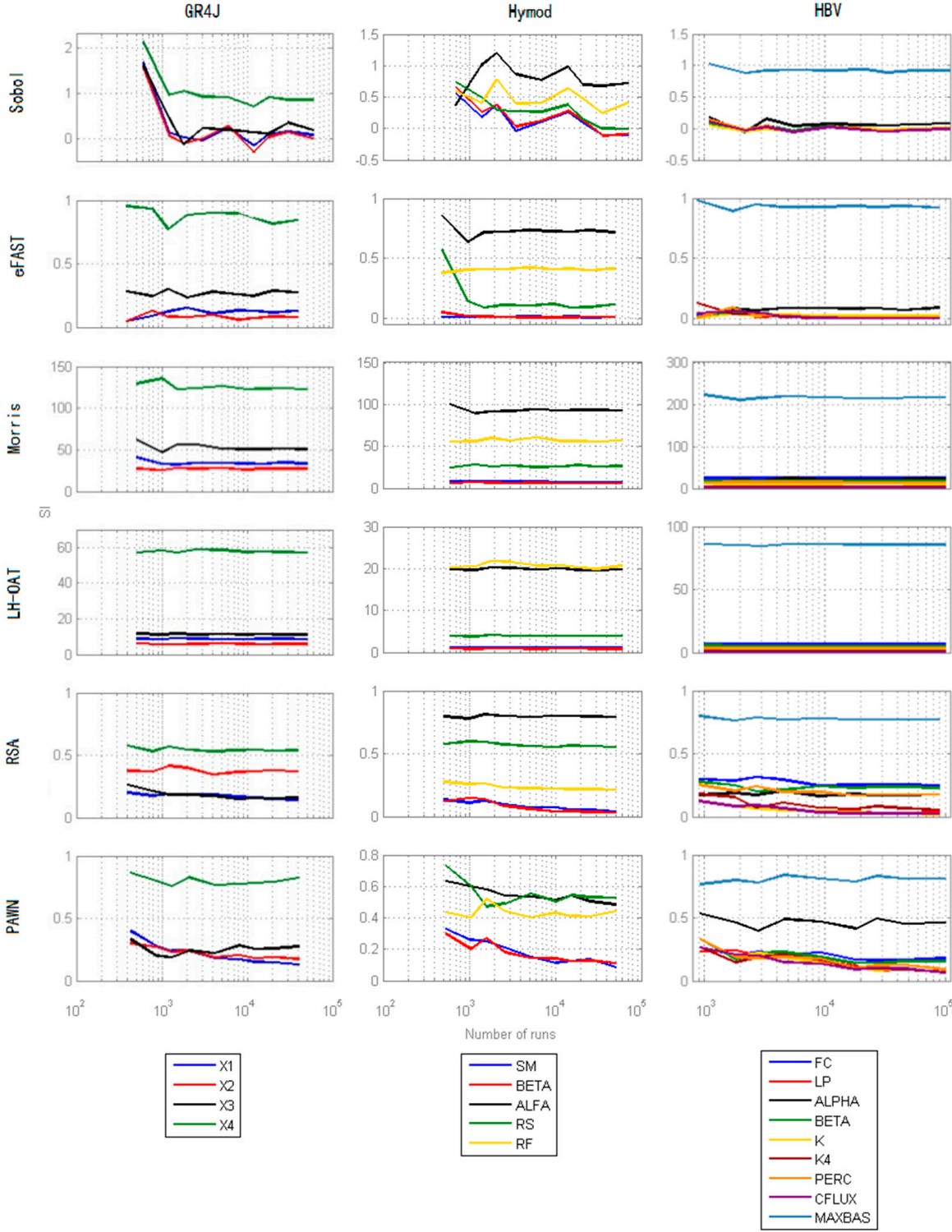

**Figure 11.** Sensitivity Indices of different Sensitivity Analysis (SA) methods with different number of runs for GR4J (**left column**), Hymod (**mid column**), and HBV (**right column**) model, the horizontal axis is in log scale.

**Table 5.** Minimum number of runs for computing Sensitivity Indices, ranking and reaching convergence for different Sensitivity Analysis (SA) method.

| Method | Minimum Number of Run for GR4J | | | Minimum Number of Run for Hymod | | | Minimum Number of Run for HBV | | |
|---|---|---|---|---|---|---|---|---|---|
| | Sensitivity Index (SI) | Rank | Conver-Gence | SI | Rank | Conver-Gence | SI | Rank | Conve-Gence |
| Sobol | 15,000 | 6000 | 60,000 | 35,000 | 2100 | 70,000 | 11,000 | 22,000 | 110,000 |
| eFAST | 1188 | 388 | - | 2485 | 485 | - | 8937 | 8937 | - |
| Morris | 1000 | 500 | 10,000 | 1200 | 600 | 18,000 | 5000 | 2000 | 20,000 |
| LH-OAT | 1000 | 500 | 10,000 | 1200 | 600 | 18,000 | 5000 | 5000 | 20,000 |
| RSA | 4000 | 400 | 8000 | 2500 | 500 | 15,000 | 10,000 | 10,000 | 30,000 |
| PAWN | 12,200 | 8200 | 40,500 | 15,200 | 10,200 | 50,500 | 18,200 | 9200 | 100,500 |

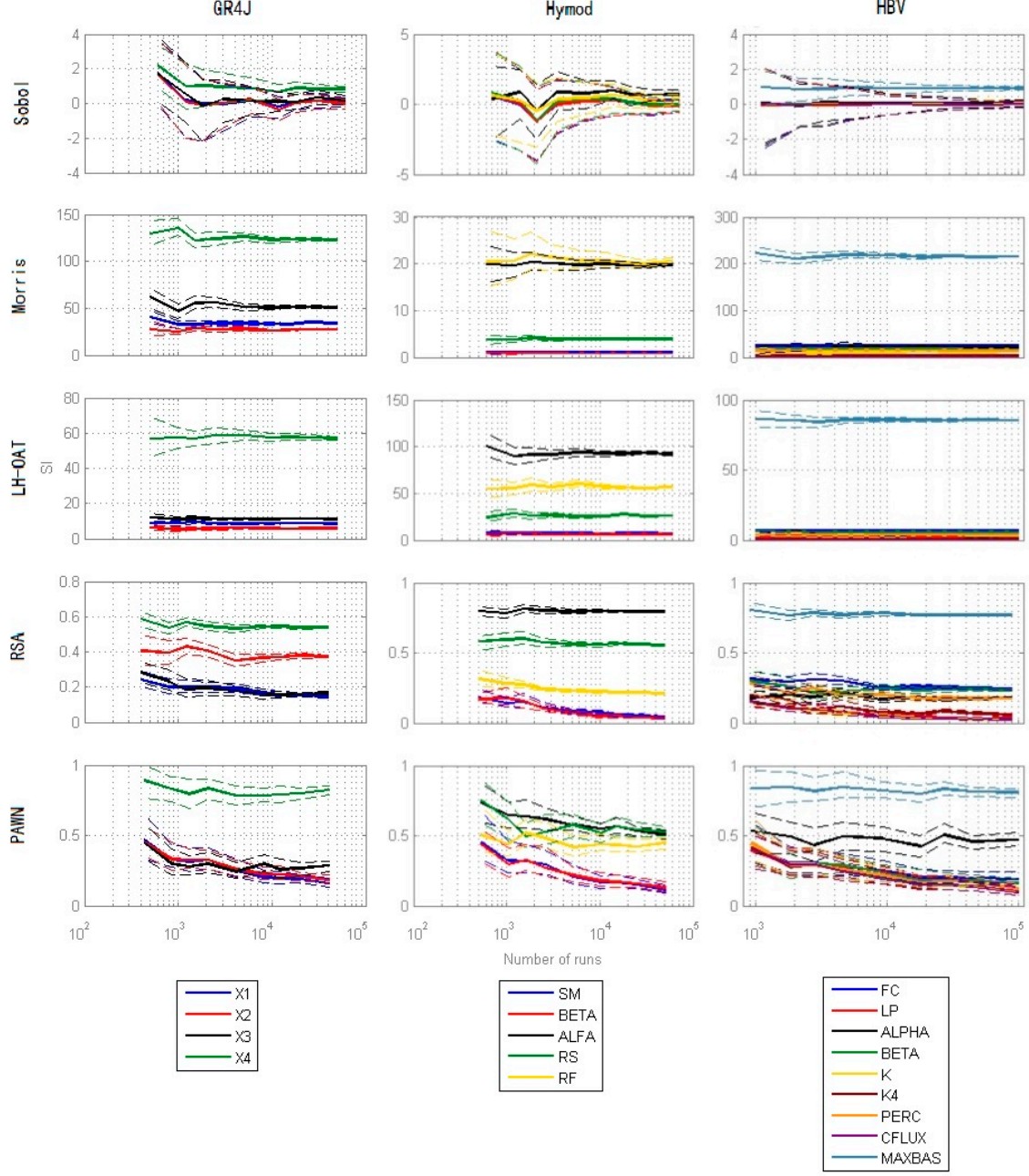

**Figure 12.** Estimate of mean and 95% Confidence Interval (CI) of different Sensitivity Analysis (SA) methods with different number of runs for GR4J (**left column**), Hymod (**mid column**) and HBV (**right column**) model, the horizontal axis is in log scale.

## 5. Discussion and Recommendations

### 5.1. Different Methods Are Based on Different Concepts

From the results above, we can see that different methods show different performances in computing SI, ranking, and convergence. One of the reasons is that there are different theories, concepts, and principles behind each method, and methods of the same category (sharing similar principles) show similar results. Comparing their performance within each category, it can be seen that GLS methods have the highest efficiency and fastest convergence speed. Variance-based and density-based methods perform less well. GLS methods use first-moments to compute SI, and the principles they use are relatively simple. Therefore, the propagation of the uncertainty in the SA is also simpler and more direct. Variance-based methods, however, with much more complex principles, result in higher uncertainty in SA. On the other hand, density-based methods may suffer from the necessity to produce reliable and accurate eCDFs, and the fact that they use K-S statistics to compute SI. As a result, they are highly unstable and uncertain. RSA performs better than PAWN, owing to its relative idea (dividing the factor vectors into only two or several sets).

Although variance-based methods seem to be less efficient in computational cost, they use more sophisticated mathematical and statistical apparatus and quantify sensitivity most accurately. Comparatively, GLS methods use only the first derivatives to compute SI, which is of course carries less information (and we can say, less accurate). Density-based methods are moment-independent, they do not need complex equations or computation to get SI, but their strength in quantifying sensitivity is problematic, as stated earlier. Generally speaking, the efficiency and depth of quantification are in inverse relationship. To obtain greater degree of quantification, it takes more model runs, and aiming to reach higher efficiency will lead to inevitable sacrifices in accuracy and reliability of the results. The method that best balances these two aspects seems to be the eFAST method. It uses variance to quantify the sensitivities, and at the same time, requires much smaller number of runs than Sobol method.

Another aspect to be mentioned is the easiness of the methods' implementation and their integration with (hydrological) models. If the method is too difficult to implement and integrate with the already existing and operational models, its use may be quite limited. This is especially true for distributed models, when sampling may be required at every grid cell, so it is not realistic to use too complex sampling methods, such as in eFAST. In these situations, methods with very simple principles like RSA and LH-OAT are more suitable.

Density-based methods seem attractive due to their simplicity, but they have two problems. On the one hand, the reliability of eCDF produced is questionable. On the other hand, the use of the Kolomogorov–Smirnov statistic to compute the Sensitivity Indices, unstable by design, may lead to slow convergence. However, they have two advantages: first, they are moment independent methods, which do not need complicated computational process; second, the results of SA can be expressed in graphs which provide yet another instrument for analysis. One of the ideas that can be explored is to quantify the results not by K-S statistics, which is the maximum difference between the two eCDFs, but to consider an integral difference (the area between two CDFs).

### 5.2. Recommendations for Choosing Sensitivity Analysis (SA) Methods

Based on the experiments and considerations presented above, we can formulate the following recommendations for choosing SA method(s):

- For simple conceptual hydrological models (not requiring much time for running them multiple times), variance-based methods as Sobol and eFAST are recommended, because they have a strong theoretical background and provide more insight into sensitivity.
- For more complex hydrological or hydraulic models that need considerable time to run, GLS methods can be used, since they are more efficient.

- For distributed models, methods with simple concepts and sampling techniques are more suitable, such as RSA and LH-OAT.
- For very complex models, e.g., 2D (or even 3D) models, like flood inundation models, or high resolution groundwater models of large aquifers, the Local SA instead of Global SA can be used [52], or LSA at a selected limited number of points in the factor (sub)space, for a reduced number of factors.
- In situations when only relative sensitivity of the factors is needed, rather than the exact value of SI, it is advisable to aim only at determining ranking or screening of SA, which needs significantly less time than the calculation of global SI.
- If time allows, it is recommended, however, to employ several different SA methods rather than using only one method.

### 5.3. Practical Framework, with the Focus on Performance Analysis

It is always useful to structure the employed sequence of steps of SA, and it is done by presenting the diagram on Figure 13. The steps in this framework are quite standard for most of SA studies, in are not very different from the framework for SA published earlier, but with the focus on the analysis of effectiveness, efficiency, and convergence of the used method(s), and a link to UA. We consider both SA and UA to be important and connected phases of model analysis, both focusing on certain aspects of model uncertainty, so it is reasonable to bring them together under one framework.

This framework assumes the model is already calibrated (as is also done in many applications of SA), however, it is also applicable to uncalibrated models for choosing a (limited) set of the (sensitive) parameters to calibrate which can improve the efficiency of calibration process.

In case there it is possible to employ several methods, we can suggest selecting one method from each category: variance-based methods, methods aggregating the local sensitivity measures, and density-based ones; the overall judgement about sensitivity will then be better informed. If time does not allow for a large number of runs, Local Sensitivity Analysis method can also be used for the calibrated or observed values of the factors.

It is also recommended to first start with a small number of sample size, and then gradually increasing the sample size until the Sensitivity Indices or ranking converges or stabilizes. (This seems to be trivial, but analysis of literature shows that is not always followed, and of course it becomes important for complex models.) The stopping criteria may vary and can be also subjective, depending on the requirement of accuracy or computational limitations; there is a balance between the accuracy of the results and the efficiency to obtain these results.

### 5.4. Limitations

There are at least the following limitations of this work, which may lead also to the future research efforts:

1. The models used in this study are only conceptual rainfall-runoff models with similar structures, so the results may be different for other types of models.
2. Evaluations of SA methods are still qualitative, so to evaluate each aspect of SA methods some more rigorous quantitative standard should be set. For example, when evaluating convergence, a threshold of the CI width should be defined for reaching convergence. Quantitative assessment will strengthen the conclusions of the comparisons.
3. Only one performance metric (RMSE) is used in this study. Parameters that is not sensitive to RMSE may affect other metrics. Various performance metrics which capture different features of model behavior should be used in the future study.

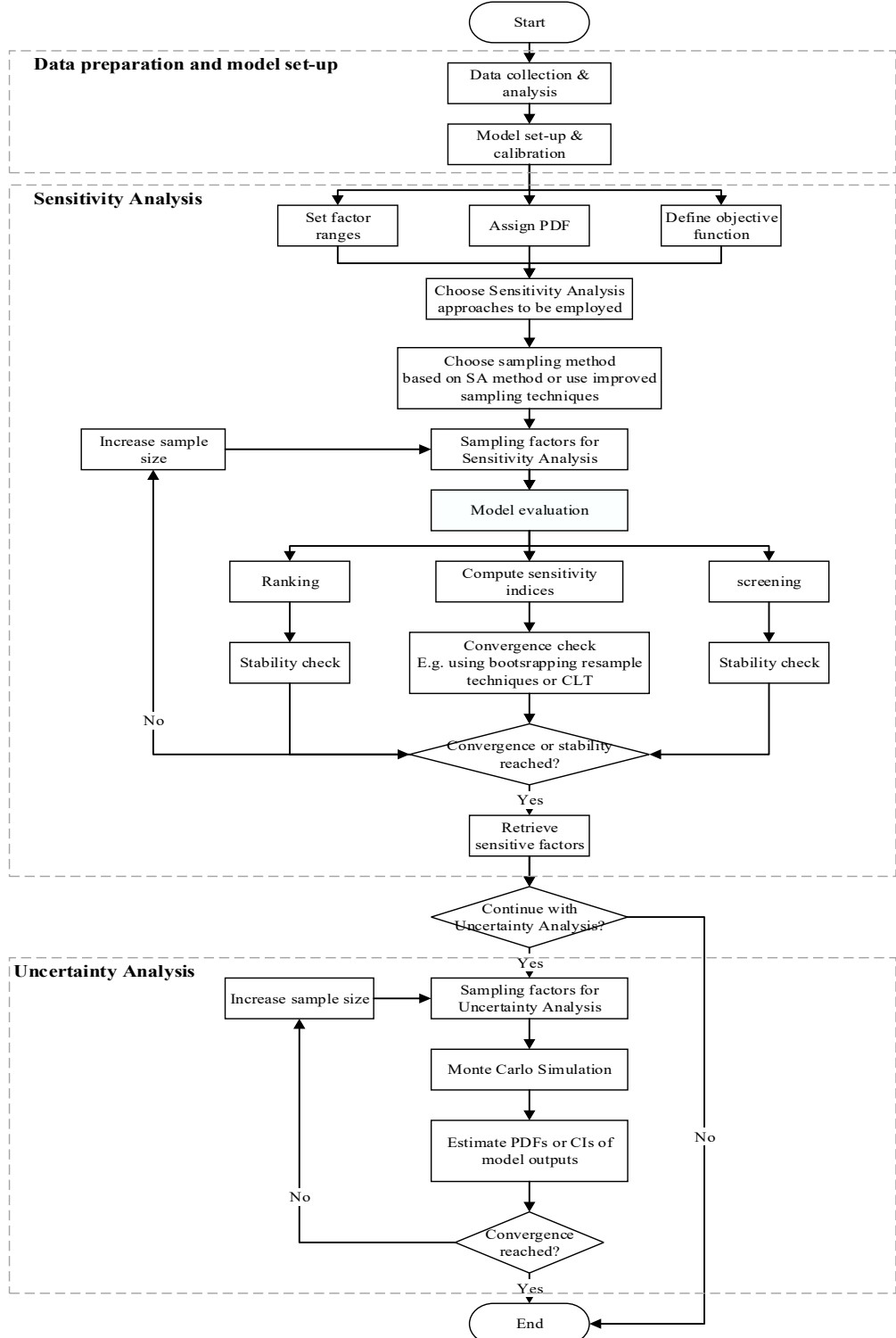

**Figure 13.** Framework for Sensitivity Analysis and Uncertainty Analysis of hydrological model.

## 6. Conclusions

SA and UA are important steps for better understanding and evaluation of hydrological models. For complex hydrological models, sample-based SA methods are often used. In this study, six different Global Sensitivity Analysis methods: Sobol, FAST, Morris, LH-OAT, RSA, and PAWN are tested on the three conceptual rainfall-runoff models: GR4J, Hymod, and HBV, with increasing complexity and

the number of parameters. The methods are compared according to the three criteria: effectiveness, efficiency, and convergence.

The results of each method are not exactly identical, but still similar to each other. All of the methods are proven to be effective. Methods from the same category show similar results as they are based on similar principles. The credibility of density-based methods is slightly undermined for two reasons: first, the reliability of eCDF produced may not be always high; second, the use of Kolomogorov–Smirnov statistic to compute the Sensitivity Indices lead to slow convergence.

The evaluation of each method's efficiency demonstrates that GLS methods, such as Morris and LH-OAT, are very efficient and stable in computing SI and ranking. Sobol method can provide quantitative results of SA, but it requires a large number of runs to obtain stable results. eFAST is much more stable and efficient than Sobol, thus it may be seen as a good alternative for the Sobol method. The efficiency of density-based methods is not so high, but RSA can give reliable results of ranking with small number of runs.

All the methods need significant number of runs (>8000) to reach convergence. The uncertainty in the values of Sensitivity Indices is not negligible. One should be careful when interpreting the results if the number of samples is not sufficiently large.

The difference in efficiency of different methods may be due to the difference in the underlying principles. Methods based on simple concepts are more efficient and stable. Methods based on the more complex concept seem to be less stable and efficient, however, their quantification of sensitivity is more accurate and reliable.

The presented recommendation for choosing SA methods, and the framework for SA and UA based on effectiveness, efficiency and convergence, as well as ease of integration with the models, add to other useful SA frameworks (workflows) (e.g., Pianosi et al., 2016 [15]), and may be of assistance for practitioners assessing reliability of their models.

Future work will be aimed at considering more SA methods (the first candidate being VARS [7,8]), developing quantitative and more informed measures for their assessment, and testing the results and recommendations against other types of models and scenarios of their practical use.

One possible avenue that can be explored further is the "multi-model approach", which is being successfully used in modelling and especially in machine learning, and which was suggested by one of reviewers of an earlier version of this paper (T. Wagener) for SA: several methods can be combined and thus potentially lead to an aggregated and perhaps more stable estimate of sensitivity. At the same time, combining methods based on different notions of sensitivity would need clear definition what type of sensitivity is then explored.

In terms of technical improvements, for calibration, SA and UA of computationally complex models, it would be useful to better aggregate the algorithms using sampling for the mentioned three stages—to be able to keep the executed model runs in one common database and smartly reuse them at various stages.

**Author Contributions:** Conceptualization, A.W. and D.P.S.; methodology and experimental set-up, A.W. and D.P.S.; software, A.W.; validation, A.W. and D.P.S.; formal analysis, A.W.; investigation, A.W.; resources, D.P.S.; data curation, A.W.; writing—original draft preparation, A.W.; writing—review and editing, D.P.S.; visualization, A.W.; supervision, D.P.S.; project administration, D.P.S.; funding acquisition, D.P.S.

**Funding:** This research was partly funded by the Russian Science Foundation, grant No. 17-77-30006, and Hydroinformatics research fund of IHE Delft.

**Acknowledgments:** The authors are grateful to the Dutch Ministry of Infrastructure and Environment for providing financial support for the first author when she followed the Master programme in Water Science and Engineering (specialization in Hydroinformatics) at the IHE Delft Institute for Water Education. The initial version of this manuscript was published as a preprint in HESS Discussions in 2018, and authors acknowledge the useful comments and suggestions of the referees.

**Conflicts of Interest:** The authors declare no conflict of interest.

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
