# Peer review of "Practical Experience of Sensitivity Analysis: Comparing Six Methods, on Three Hydrological Models, with Three Performance Criteria"

_water, doi:10.3390/w11051062_

Round 1
Reviewer 1 Report
The paper focuses on comparison of sensitivity analysis methods for hydrological models. The paper contains interesting results with detailed explanation of the methods, but without details of the hydrological model utilized. The readers may want to know details of the hydrological models. I strongly recommend the authors to add an appendix that describe full details of the models, even if they are long, for the completeness of the paper.
Author Response
please see attached pdf file.

Reviewer 2 Report
I read the paper „Practical experience of sensitivity analysis: Comparing six methods on three hydrological models, with three performance criteria” with interest. I think that understanding the differences between sensitivity analyses is important for making good decisions when having to implement one and for understanding the limitations of a given model.
In my opinion, the main aspects that need to be improved are the description of the way the study was set up and the English language. I also do have comments regarding the results and discussion. I think this paper still requires quite an amount of work. I therefore recommend a major revision or a rejection, which would give the authors more time to improve their paper.
Comments in order of appearance:
- I would not include the reference to the Dapoling-Wangjiaba catchment in the abstract, as those results are not presented in the paper
- Lines 115-117 refer to qualitative and quantitative SA. It would be nice to have a definition for these terms.
- In Lines 109-120 the authors refer to studies comparing sensitivity analysis methods and afterwards to review papers about SA. Around line 120 they highlight that the review papers to not compare sensitivity methods. I found this part somehow confusing.
- Lines 121-124 are about the uncertainty of sensitivity analyses, as well as lines 135-138. I also found this confusing.
- The global SA methods are presented in chapter 2. The Generalized sensitivity analysis (GSA) method is presented in section 2.1.1 and section 2.1.4 is about density methods. Isn’t GSA also a density method?
- The Kolgomorov-Smirnov statistic is discussed in detail in the results and discussion sections. I think it should be described with more detail.
- In line 235 it is specifies that derivation of empirical PDFs is either too simple, resulting in inaccurate results (note that simple approaches are not necessarily less accurate than more complex approaches) or too complex to implement. Do you have an example of an approach for calculating empirical PDFs that is too complex to implement?
- Section 2.1.5 is about meta-modelling. It is presented as a GSA method. I am not sure if this is a sensitivity analysis method. I see it more as an approach getting the required model runs in less time. These model runs then need to be “processed” by a sensitivity analysis method (e.g. Morris, Sobol).
- In section 3.1.1 the concepts of efficiency and convergence of the SA methods are introduced. I do not really understand clearly the difference between these criteria. I understood that more efficient SA models require fewer model runs as they converge faster. Could you clarify this?
- Line 279-280 says that the SA methods will be evaluated regarding their ability for supporting screening, ranking and sensitivity indices values. Maybe it would be helpful to include a definition of these terms? It is not explained how it was decided that a model is good for ranking/screening/estimating sensitivity values by looking at the dotty plots references.
- Section 3.1.3: I do not understand what is meant by “base sample set”. Is this the total number of model runs that was carried out? Is the benchmark test run with a part (which part?) of this base sample test? It says that the base sample sizes varies for the different methods. Can you give the size used for each method? How is the minimum sample size found? Line 289 says that the sample sized was found for the ranking and the sensitivity indices. Why was screening not considered?
- Section 3.1.4: Which are the statistics of the sensitivity indices distribution that were calculated?
- Line 365-367: It says that the resample size for evaluation is 100. What does this mean?
- Table 4: I think it is necessary to give more information about the measures used. They are needed for understanding Figure 8. There are some measures that are not mentioned in the text. What is the Effect S, the modified mean of effect elementary ?
- Table 4: I do not understand what the benchmark run is and how the evaluation with the base samples works
- Figure 6: Some plots have a wrong label (SM, RS, RF) as these are parameters of the Hymod model.
- In lines 411-415: it says that the results for the PAWN and RSA methods are different to the results of other methods because the use of the K-S distance brings instability into the results. I am a bit confused by this. I think table 7 shows the results for the benchmark – which I assume means that the estimated already converged? - so the estimates should be stable.
- What is meant by “eCDF”? in lines 439, 440, 518, 524
- Figure 8, can you please add a label for the y-axis?
- Table 5: it shows the number of model evaluation needed for screening, ranking and estimating the sensitivity indices. How were these number obtained? In some cases they are larger than 100.000. Was the model run so many times?
- Line 505-506: I liked this statement, that the efficiency and depth of the analysis are in an inverse relationship.
- Section 5.3: shows a framework for sensitivity and uncertainty analyses. I am not sure how this relates to the previous sections. Maybe it can be better linked to the remaining study.
- Figure 10: I do not think that the flowchart part with the uncertainty relates well to the remaining study. The flowchart brings up many aspects that are not mentioned in the text. For example, the setting of the factor ranges, the assignation of a PDF.
- Section 5.4: I think that another limitation of the study is that is only compares one performance metric (RMSE) which is rather sensitive to high flows. Other metrics (e.g. bias, correlation, mean absolute error) might show that parameters not affecting the RMSE do affect other metrics (see van Werkhoven et al., 2009).
References
van Werkhoven K.; Wagener T.; Reed P.; Tang Y. 2009. Sensitivity-guided reduction of parametric dimensionality for multi-objective calibration of watershed models. Advances in Water Resources 32(8):1154-1169, doi:10.1016/j.advwatres.2009.03.002.
Author Response
please see attached pdf file.

Round 2
Reviewer 2 Report
I think the paper improved a lot.
I do have a very minor comment:
*** I agree that there is a definition of empirical CDFs is given in line 172 and 183. What is missing on my opinion is "(eCDF)" after the definition, so that readers knnow that eCDF stands for empirical CFD